# Cytokine Landscape in Central Nervous System Metastases

**DOI:** 10.3390/biomedicines10071537

**Published:** 2022-06-28

**Authors:** Julie Marin, Fabrice Journe, Ghanem E. Ghanem, Ahmad Awada, Nadège Kindt

**Affiliations:** 1Laboratory of Clinical and Experimental Oncology (LOCE), Institut Jules Bordet, Université Libre de Bruxelles (ULB), 1070 Brussels, Belgium; julie.marin@bordet.be (J.M.); fabrice.journe@bordet.be (F.J.); gghanem@bordet.be (G.E.G.); ahmad.awada@bordet.be (A.A.); 2Laboratory of Human Anatomy and Experimental Oncology, Institut Santé, Université de Mons (UMons), 7000 Mons, Belgium; 3Department of Medical Oncology, Institut Jules Bordet, Université Libre de Bruxelles (ULB), 1070 Brussels, Belgium

**Keywords:** brain metastases, leptomeningeal metastases, microenvironment, reactive astrocytes, microglia and cytokines

## Abstract

The central nervous system is the location of metastases in more than 40% of patients with lung cancer, breast cancer and melanoma. These metastases are associated with one of the poorest prognoses in advanced cancer patients, mainly due to the lack of effective treatments. In this review, we explore the involvement of cytokines, including interleukins and chemokines, during the development of brain and leptomeningeal metastases from the epithelial-to-mesenchymal cell transition and blood–brain barrier extravasation to the interaction between cancer cells and cells from the brain microenvironment, including astrocytes and microglia. Furthermore, the role of the gut–brain axis on cytokine release during this process will also be addressed.

## 1. Introduction

Central nervous system (CNS) metastases are localized in brain parenchyma and leptomeninges. Brain metastases (BM) are considered as the most common malignancy of the brain in adults and may appear years or decades after the diagnosis of the primary cancer or may be the first manifestation of a previous undiagnosed cancer. BM occur in more than 40% of patients with lung cancer, breast cancer and melanoma mainly due to the improvement of systemic control of the primary cancer and/or of the visceral metastases [1,2,3]. Beside the origin of the tumor, other factors affect the development of BM e.g., extracranial metastases, molecular subtype, gender, age, ethnicity and geographical location of the patients [4,5]. The overall survival of treatment-naïve patients diagnosed with BM ranges from 1 to 3 months. For patients treated with surgery, whole brain radiation therapy, chemotherapy or combinations, their survival can be extended to 4–11 months [6]. These patients have a worse prognosis due to poor treatment efficacy and frequent local relapse of BM after resection of the tumor [7,8].

Leptomeningeal metastases (LM) appear when tumor cells infiltrate the leptomeninges of the brain, spinal cord, as well as the cerebrospinal fluid (CSF) [4]. They occur with a higher frequency in melanoma (23%), lung cancer (9–25%) and breast cancer patients (5%), while the incidence for hematological malignancies is 5 to 15% [9,10]. Cancer cells reach the leptomeninges by several ways, e.g., dissemination through the arterial or venous circulation, or endoneural or perineural spread [11]. Brain tumor neurosurgery is a risk factor promoting the entry of cancer cells in the ventricular system [12]. In the absence of BM, the most frequent risk of dissemination is the present of bone metastases in vertebra. Cancer cells may access the bones neighboring the CNS via venous or arterial vessels and then invade leptomeninges by direct extension or spread along the perivascular spaces of the vertebral veins [11]. Moreover, when the tumor is located in the leptomeninges, the cancer cells have to adapt to a new microenvironment that is totally different from the one in the primary tumor [13].

Inflammation is the major cause of metastasis development, which involves small chemical mediators called cytokines, chemokines and growth factors that are secreted by cancer cells themselves and cells from tumor environment such as immune cells and fibroblasts [14,15,16]. Some of these cytokines can be detected in the circulation or other biofluids and thus have the potential to be biomarkers for detecting cancer, predicting disease progression and managing therapeutic choices. Furthermore, some studies have also shown that changes in the expression of pro- and anti-inflammatory cytokines are associated with carcinogenicity of cells and tumor progression in solid cancers. It is therefore essential to better understand the cytokine–tumor cell interactions in order to search for new biomarkers of diagnosis and response to immunotherapy [14,15,17]. In this review, we will discuss the role of cytokines during BM development and progression.

## 2. Development of Brain and Leptomeningeal Metastases

Brain and leptomeningeal metastases occur through cancer cells dissemination by systemic routes and crossing the blood–brain barrier (BBB) or blood–CSF barrier (B-CSF) (Figure 1). This is favored by an epithelial-to-mesenchymal cell transition (EMT), affecting cell-to-cell adhesion by the gradual loss of E-cadherin. During cancer cell invasion, inflammatory cytokines, mainly TGFβ, TNFα and IL-6, activate transcription factors such as Smads, NF-κB and Snail, driving EMT forward. In addition, immune cells and other cells of the tumor microenvironment (TME), such as cancer-associated fibroblasts, contribute to EMT by the release of vascular growth factors and matrix-degrading enzymes [18]. Once in the bloodstream, cancer cells escape the circulation by extravasation in a new site and colonize target organs [19]. In order to cross the BBB, which is composed of endothelial cells surrounded by pericytes and astrocytic endfeet, cancer cells cooperate with activated glial cells to alter its permeability [5]. IL-1β raises the permeability of the BBB, as shown by transmigration assays using a BBB-on-a-chip where triple negative breast cancer cells (MDA-MB-231) transmigration increased when the device was pretreated with this cytokine [20]. In addition, it was demonstrated in vivo that the development of breast cancer brain metastases (BCBM) could be favor by a truncated form of glioma-associated oncogene homolog 1 (TGLI1). This transcription factor can upregulate vascular endothelial growth factor-A (VEGF-A), VEGF-C, CD24, and CD44, leading to increased angiogenic potential, growth, migration and invasion of breast cancer cells [21]. Syndecan-1 was also reported to contribute to BCBM both in vivo and in vitro. Indeed, its overexpression is associated with increased BM and facilitates cancer cell transmigration across the BBB due to changes in breast cancer cell-secreted cytokines and chemokines such as IL-6, IL-8 and CCL5 [22]. In BCBM patients, it has been shown that the serum levels of fractalkine (CX3CL1) and CXCL13 are significantly increased, leading to the alteration of the BBB permeability in vitro [23]. Moreover, it was illustrated that breast cancer cell interactions with brain endothelial cells and BBB extravasation are promoted by YTHDF3, a N6-methyladenosine (m6A) “reader” protein that enhances the translation of ST6 N-Acetylgalactosaminide Alpha-2,6-Sialyltransferase 5 (ST6GALNAC5), Gap Junction Protein Alpha 1 (GJA1) and Epidermal Growth Factor Receptor (EGFR) mRNA [24]. In lung cancer, Liu et al. have demonstrated that the level of the adipokine visfatin in the serum of BM patients increased significantly, but also that visfatin promotes cancer cell migration across BBB, in vitro, associated with the upregulation of the CC chemokine ligand 2 (CCL2) [25]. Likewise, lung cancer cell lines release factors such as TNFα and VEGF-A which enhance E-selectin expression on endothelial cells of BBB and thus increase the adhesion of cancer cells on these endothelial cells [26]. Tumor-derived exosomes can also play an important role in the development of BM. Indeed, high Cell Migration Inducing Hyaluronidase 1 positive (CEMIP) exosomes coming from diverse cancer cell types induce pro-inflammatory vascular niche by upregulating cytokines and chemokines such as TNF, Prostaglandin-Endoperoxide Synthase 2 (PTGS2) and CCL in the microglia, leading to BBB dysfunction [27]. Furthermore, in non-small cell lung cancer, TGFβ1 mediates the production of exosomes enriched with lnc-MMP2-2, a long non-coding RNA, which promote the destruction of tight junction leading to an increase of the BBB permeability [28]. Glial cells, such as astrocytes, can also induce cancer cell transmigration through the BBB, notably by the secretion of CCL2 as demonstrated in a 3D in vitro human BBB model. In fact, breast cancer cells (MDA-MB-231) express CCR2, which interacts with CCL2 to mediate cancer cell extravasation into the brain [29]. The same observation was made in a melanoma BM model where astrocytes secrete CXCL10, promoting migration of melanoma BM cells expressing CXCR3, the CXCL10 receptor. These data are confirmed by the decrease of BM formation after the down-expression of CXCR3 in melanoma cells [30].

Regarding LM, the leptomeningeal space is separated from the systemic circulation by the choroid plexus producing the CSF. Hence, cancer cells may cross the blood–CSF barrier constituted of choroid plexus epithelial cells [31] (Figure 1). So far, few studies have explored the molecular and genetic mechanisms involved in LM development. Boire et al. highlighted the role of the innate immunity mediator complement 3 (C3) in cancer cells that metastasize to the leptomeninges [32]. Their experiments display that C3 is released from cancer cells to interact with C3a receptor expressed on the surface of choroid plexus cells, inducing the disruption of the choroidal B-CSF barrier, allowing the entry of growth factors and cytokines in the CSF space, and conducting tumor growth in leptomeninges [32].

**Figure 1 biomedicines-10-01537-f001:**
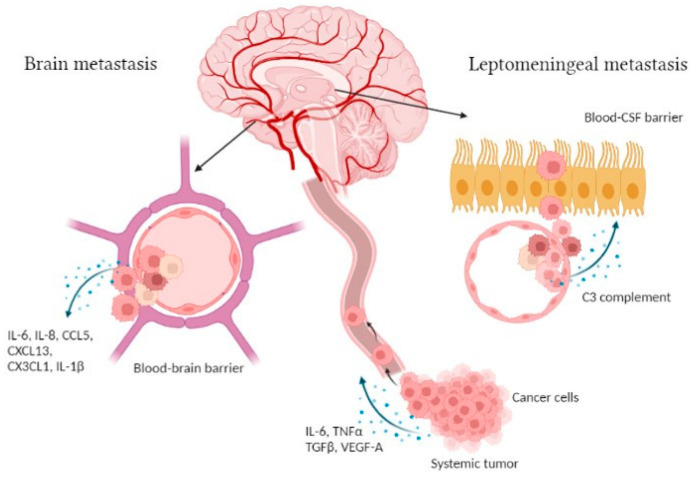
Illustration of the brain and leptomeningeal metastasis development. During the first steps of the CNS metastasis process, cells dissociate from the systemic tumor site and secrete various factors, such as IL-6, TNFα, TGFβ and VEGF-A [18]. Those cells travel in the bloodstream to the brain parenchyma where they cross the BBB following cytokine release, i.e., IL-6, IL-8, CCL5, CXCL13, CX3CL1 and IL-1β [20,22,23]. Regarding leptomeningeal metastasis, cancer cells pass through the B-CSF barrier via C3 complement secretion [32]. Figure created with Biorender.

### 2.1. Brain Metastases Establishment and Progression

In triple negative breast cancer (TNBC), gene expression profiling has revealed that the chemokine CXCL8 in tissue samples is associated with a poor prognosis and possibly associated with BM, but also increases cell invasion using TNBC cell lines [33]. This chemokine has also been highlighted by another genomic analysis where 27 hub genes were specifically identified from BCBM cells, including EGF, CXCL8, MMP9 and CXCL16, known to be implicated in tumor progression [34]. Moreover, the authors discovered that the expression of *CXCR4* gene is overexpressed compared to breast cancer lung metastasis cells, with CXCR4 interacting with CXCL12 to induce breast cancer cell migration [34,35]. Furthermore, CXCL1 was significantly increased in BM tumor cells from patients with breast cancer. CXCL1 is the ligand of CXCR2, which is the main chemokine receptor that increases in neutrophils when cultured with conditioned medium from BCBM cells. Thus, CXCR2 activation stimulates the recruitment of tumor-associated neutrophils into BM [36]. In addition, TGFβ2, a member of the TGFβ cytokine family, plays an important role for BM establishment as demonstrated in a metastasis mouse model, where melanoma cell lines expressing high levels of TGFβ2 grew only in the brain parenchyma [37]. In 2019, another study identified three genes that are downregulated differentially in lung biopsies of patients with lung cancer BM compared to patients without BM; these genes are CD37 (expressed on immune cells), cystatin A (a protease inhibitor) and IL-23A [38]. Of note, the latter interleukin is a pro-inflammatory cytokine that induces cell proliferation at a low level but inhibits cell proliferation at a high level in a lung squamous cell carcinoma model [39]. On the other hand, once cancer cells cross the BBB, they interact with cells from brain parenchyma as astrocytes and microglia, notably via the release of cytokines (Figure 2). Undoubtedly, the crosstalk between the cancer cells and such brain cells are of importance in the process of BM development.

#### 2.1.1. Role of Reactive Astrocytes

Astrocytes are one of the major components within the brain microenvironment that perform many functions in the maintenance of brain homeostasis and tissue repair processes. Therefore, metastatic cancer cells that preferentially grow in the brain have to find strategies to adapt and favorably interact with these unfamiliar cell players [40]. Astrocytes undergo transcriptomic reprogramming into reactive astrocytes following stimuli as danger-associated molecular patterns (DAMPs) or pathogen-associated molecular patterns (PAMPs) [41]. Metastatic cancer cells established in the brain switch astrocytes into a pro-tumoral phenotype, via the release of IL-1β, and create functional gap junctions with astrocytes in the microenvironment, allowing the construction of a channel for bidirectional communication to support tumor progression [42,43]. Through these gap junctions, 2′3′-cyclic GMP-AMP (cGAMP) is secreted by cancer cells to reprogram astrocytes that give rise to a pro-inflammatory program, characterized by the production of a variety of cytokines (e.g., IFNα and TNF). These cytokines support brain metastatic cell proliferation by activating STAT1 and NF-κB signaling [43]. Reactive astrocytes associated with BM are also charactrized by STAT3 activation due to the BM cell release of several cytokines as macrophage migration inhibitory factor (MIF) and TGFα. This activation induces tumor progression, notably by decreasing CD8^+^ T cell infiltration in the brain microenvironment [44]. In glioblastoma, it was demonstrated that purified astrocytes from tumor specimens showed a reactive state which is marked by response to IL-10 and IFNγ resulting in an activation of the JAK/STAT pathway [45]. In BCBM cells (MDA-MB-231BrM), there is an upregulation of COX2 and prostaglandin, which are able to activate astrocytes and induce the release of CC chemokine ligand 7 (CCL7) by those activated glial cells [46]. Astrocytes can also modulate the blood tumor barrier (BTB). Indeed, in BCBM it has been shown that astrocytic sphingosine-1 phosphate receptor (S1P3) is upregulated and mediates its effects via IL-6 and CCL2 astrocyte secretion, which increases BTB permeability [47]. Furthermore, CXCL1, TNFα, TGFα and IFNα secretions are increased by astrocytes co-cultured with brain metastatic cancer cells. These cytokines induce the expression of S100A9 in cancer cells, leading to an inflammatory response and radioresistance [48]. Finally, micro-RNAs (miRNA), which are small non-coding RNAs that post-transcriptionally control expression of genes by targeting mRNA, can be associated with cancer progression and metastases [49]. In BCBM, it was demonstrated that miRNA-1290 and miRNA-1246, derived from extracellular vesicles secreted from breast cancer cells, induced a strong activation of astrocytes via the suppression of expression of the transcriptional repressor FOXA2, leading to the secretion of the Ciliary Neurotrophic Factor (CNTF). Moreover, astrocytes that overexpress miRNA-1290 increase the ability of breast cancer cells to grow and colonize the brain [50].

#### 2.1.2. Role of Microglia/Macrophage

Brain TME is also composed of distinct macrophage populations, including both tissue-resident microglia and bone marrow-derived macrophages (BMDMs) [40]. Microglia become active in pathological conditions as cancer, and release abundant factors of inflammation that allow tumor progression [51]. On the other hand, under pathological conditions, circulating monocytes are also recruited to the brain to become BMDMs and contribute to tumor progression depending on phenotypes [40]. Indeed, microglia as well as BMDMs can polarize into pro-inflammatory (M1) or pro-tumoral phenotype (M2) depending on the pathological state [52]. Interestingly, Wu et al. showed that the role of nicotine in the incidence of lung cancer BM may be explained by the polarization of microglia into an immunosuppressive M2 phenotype [53]. The induction of M2 microglia polarization by nicotine occurs via an increased expression of α4β2 nicotinic acetylcholine receptor in microglia, which is the most sensitive receptor of nicotine in the brain. The interaction between nicotine and its receptor induces the activation of the JAK/STAT3 pathway, leading to the release of IGF-1 and CCL20 from M2 microglia cells that further stimulate tumor cell growth and cancer cell stemness [53]. Of note, estrogens can also polarize microglia to the M2 phenotype in BM from breast cancer. The induction of this phenotype is mediated by the activation of STAT3, which allows the increased secretion of CCL5 and decreased secretion of TNFα leading to tumor growth. This effect of estrogens is reversed by the use of tamoxifen, which is known to reduce the incidence of BCBM [54]. In lung cancer brain metastatic cells, Jin et al. have identified IL-6 as a key regulator that promotes anti-inflammatory/pro-tumoral microglia via JAK2/STAT3 signaling pathway. Moreover, they demonstrated that there is a higher level of IL-6 in the serum of patients with BM [55]. Furthermore, myeloid cells around and in BM downregulate the CX3CR1 gene, leading to tumor progression, CXCL10 upregulation and recruitment of VISTA^hi^ PD-L1^+^ CNS-myeloid cells that suppress T cell infiltration. This was confirmed by the combination of αVISTA and αPD-L1 antibodies, which increased the infiltration of CD3+ T cells [56]. Similarly, BMDMs showed an important recruitment of anti-inflammatory/pro-tumoral macrophages (arginase 1+(Arg1^+^)) over time in a mouse model of BCBM [57]. Finally, exosomes released by lung cancer cells and internalized by brain endothelial cells can also send a suppressive message to microglia, resulting in less M1 and more M2 phenotype microglia. This was confirmed by the use of an inhibitor of exosome release, GW4869, leading to an increase in IL-1β, a marker of M1 cells, and a decrease in CD206, an M2 cell marker [58]. As for astrocytes, miRNAs (miR196a-5p) derived from extracellular vesicles secreted by cancer cells can also impact the activity of microglia. Indeed, extracellular vesicle release by nasopharyngeal carcinoma increase microglia proliferation and levels of IL-6, IL-8, CXCL1 and TGFβ [59].

Emerging evidence has highlighted the presence of a crosstalk between astrocytes and microglia via the secretion of various mediators such as cytokines, growth factors and neurotransmitters [60]. In microglia/astrocytes co-culture, cytokines such as IFN-γ, IL-10, GM-CSF and others are more elevated compared to microglia or astrocyte culture alone, enhancing the proliferation of microglial cells [61]. In addition, during neuroinflammation, the secretions of IL-1α and TNFα by mouse microglia induce a pro-inflammatory state in astrocytes that then secrete neurotoxins driving neuronal death [62]. Regarding BM, it was demonstrated that beta amyloid secreted by melanoma cells activates surrounding astrocytes into an anti-inflammatory phenotype and prevents phagocytosis of melanoma by microglia [63]. Finally, it has been reported that pSTAT3^+^ reactive astrocytes release a high level of MIF, the ligand of the CD74 receptor expressed on macrophage/microglia. In this context, CD74^+^ macrophage/microglia infiltration is increased in BM while CD74 in the nucleus acts as a transcription factor and induces the expression of midkine, a pro-tumorigenic molecule stimulating tumor growth [44].

**Figure 2 biomedicines-10-01537-f002:**
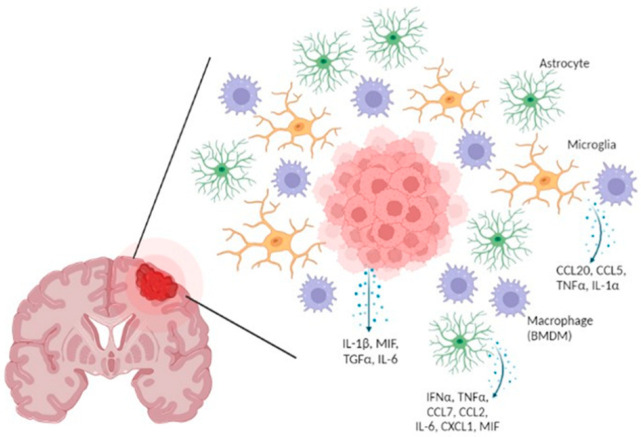
Pattern of cytokines which are released during brain metastasis establishment. Cancer cells (red) release several cytokines such as IL-6, IL-1β, TGFα and MIF in the brain leading to the activation of astrocytes (green) and macrophages (violet)/microglia (yellow) that secrete multiple cytokines and chemokines to support tumor growth [43,45,47,48,49,52,53,54]. Figure created with Biorender.

#### 2.1.3. Role of Other Immune Cells

Other immune cells such as lymphocytes and neutrophils are involved in the progression of BM. The function of lymphocytes and neutrophils in inflammatory responses is well known, but, for the latter, their role in tumor progression and metastasis development remains unclear. Regarding lymphocytes, their infiltration is significantly lower in BM than in primary tumor, as seen in breast and lung cancers [64,65,66]. BM patients with lower tumor-infiltrating lymphocytes (TILs) display a poorer prognosis compared to BM patients with a higher number of TILs. Moreover, within the group of patients who present fewer TILs, VISTA expression is higher, which could explain the decreased infiltration of lymphocytes as observed by Guldner et al. [56,67]. In addition, it was demonstrated by Mustafa et al. that T cells increase the ability of breast cancer cells to cross the BBB thanks to the expression of Guanylate Binding Protein 1 [68]. On the other hand, regulatory T cells accumulate in BM and release IL-10, IL-4 and IL-13, which may induce the development of TAM and the suppression of cytotoxic CD8 T cell responses, leading to an immunosuppressive microenvironment [69].

Regarding neutrophils, initially, the maturation of neutrophil precursors into mature ones is promoted by the cytokine Granulocyte-Colony Stimulating Factor (G-CSF), which is also responsible for the increase in the number of neutrophils in the blood. During neutrophil recruitment, IL-8 will play a crucial role in addition to the protective inflammation at the site of metastasis [70]. New genetic tools have recently been developed to highlight the importance of neutrophils in the TME, called tumor-associated neutrophils (TANs), particularly in tumor progression and resistance [71,72]. TANs are phenotypically very heterogeneous and therefore have several subpopulations with different functions. The presence of Transforming Growth Factor-β (TGFβ) promotes polarization towards a pro-tumor phenotype (N2), whereas the presence of IFNβ or inhibition of TGFβ signaling will conversely favor the anti-tumor phenotype (N1) [73,74]. Nowadays, studies have identified a particular subgroup with an immunosuppressive function that is recruited specifically to the brain to promote the development of metastasis. In BM, cancer cells overexpress an epigenetic modification protein, Enhancer of Zeste Homolog 2 (EZH2), and change its function from a methyltransferase to a transcription factor that increases the expression of c-JUN. The transcription factor c-JUN then upregulates some pro-tumorigenic inflammatory cytokines such as the G-CSF. This cytokine will play a key role in the recruitment of Arg1-positive and PD-L1-positive immunosuppressive neutrophils to the brain to support metastasis growth [73,75]. Other chemoattractants such as CXCL5, CXCL8 and CXCL17 are upregulated in BM and also play a role in immunosuppressive neutrophil recruitment into the metastatic niche [75,76]. In recent years, researchers have identified a very interesting prognostic factor based on neutrophil and lymphocyte counts. Indeed, in patients with BM, a high neutrophils-to-lymphocytes ratio (NLR) in the peripheral blood has been identified as a biomarker of poor prognosis [73,74,75,77]. Increased NLR was associated with a rise of peritumoral macrophage infiltration and increased expression of IL-6, IL-7, IL-8, IL-9, IL-12, IL-17 and IFNγ [74]. Finally, one study has revealed the role of the CXCR2 receptor in neutrophil behavior in BCBM. Upregulation of the chemokines CXCL1, CXCL8 or IL-8 in BCBM cells dynamically activates CXCR2 receptors on neutrophils. In response to ligand overexpression, CXCR2 in neutrophils is upregulated, resulting in the modification of certain functional neutrophil responses. Using neutrophil–tumor co-cultures, researchers have demonstrated that upregulation of CXCR2 increases the recruitment of TANs to enable the formation of neutrophil extracellular traps (NETs) in response to pro-inflammatory stimuli. These NETs appear to promote metastasis by enhancing tumor cell seeding and colonization in host organs. Based on these results, authors hypothesized that CXCR2 activation could be used by metastatic tumors as a mechanism to delay apoptosis and to program tumor-infiltrating NETs into a pro-NETotic state, to promote subsequent migration and invasion of metastatic tumor cells [36].

### 2.2. Leptomeningeal Metastases Establishment and Progression

Once cancer cells have passed through the B-CSF barrier, they are faced with substantial microenvironmental challenges, including inflammation and sparse micronutrients. To explore the mechanism by which leptomeningeal metastasis cells overcome these constraints, Chi et al. examined CSF from five patients with LM by single-cell RNA sequencing. They found that cancer cells in the CSF express the iron-binding protein lipocalin-2 (LCN2) and its receptor SCL22A17. The sequencing of macrophages from the CSF showed high expression levels of transcripts downstream of JAK/STAT and NF-κB promoters with high concentrations of IL-6, IL-8 and IL-1β in the CSF of LM patients [78]. In vivo, they demonstrated that these cytokine-derived macrophages induce LCN2 expression by cancer cells. The LCN2/SLC22A17 system is a high affinity iron system that supports cancer cell growth and is inhibited by iron chelation therapy [78]. Moreover, it was demonstrated by the Jandial group that the cytokine Granulocyte-Macrophage Colony Stimulating Factor (GM-CSF) plays a major role in the establishment of LM from HER2+ breast cancer. Indeed, when they inhibited GM-CSF with anti-GM-CSF antibodies combined with pan-Aurora kinase inhibitor (CCT137690), they reduced HER2+ LM cell growth in vivo [79]. Similarly, when they used the JIB04 treatment, which is a selective inhibitor of Jumonji demethylases overexpressed in their LM cell lines, they observed a downregulation of GM-CSF expression that prevents cancer cell proliferation [80].

## 3. Role of Cerebrospinal Liquid (CSF)

CSF produced in the ventricles by the choroid plexus is an important sampling source to understand CNS malignancies. Initially, CSF was considered useful only for maintaining the homeostatic environment necessary for normal brain function. However, in addition to providing an homeostatic environment, CSF is a pathway for signal transmission during neurological development and progression of primary and metastatic brain tumors [81,82]. Researchers hypothesized that a group of cytokines and chemokines called immunokines can be detected in the CSF and thus can help as biomarkers in understanding the course of BM. In melanoma BM patients, the CSF levels of IL-1α, IL-4, IL-5 and CCL22 significantly decrease while those of CXCL10, CCL4 and CCL17 significantly increase compared to non-disease group [82]. Clustering of patients based on their immunokine profiles allowed the discrimination of groups according to their health status. Therefore, these data suggest that the relationship between immune system and melanoma may be used as a prognostic biomarker to evaluate patient outcome. Finally, CSF immunokine profiles could also serve as diagnostic biomarkers for the detection of melanoma BM [82]. In 2020, a research group from Singapore investigated the involvement of CSF cytokines in a small cohort of medulloblastoma patients with or without intracranial/spinal metastases to better understand the implication of inflammation in the development of metastasis. Following a proteomic analysis (proteome array blot), they found a higher release of CCL2 into the CSF in the metastatic group compared to the non-metastatic group. Hence, CCL2 could be used as a new biomarker for metastatic medulloblastoma patients [83].

## 4. Role of Microbiota in Central Nervous System Cancer Development

During the last 10 years, research on the involvement of microbiota in tumor progression has exponentially increased from 58 articles in 2011 to 1240 articles in 2021 (PubMed) and now appears as a hallmark of cancer [84]. Indeed, dysbiosis can induce tumor-promoting inflammation and genomic instability as demonstrated by Li et al., where increasing secretion of cathepsin K (CTSK) by gut microbiota influenced macrophage polarization into M2 tumor-associated macrophages in colorectal cancer with the increase in IL-4, IL-10 and IL-17 release, leading to a metastatic phenotype [85]. Regarding the brain, the relationship between the gut–brain axis during brain cancer/metastasis development has recently been investigated and some evidence has been reported. Indeed, the gut microbiota affects brain function and behavior through endocrine, neural and immune pathways [86]. Some bacterial phyla metabolize amino acids such as arginine and tryptophan that produce, for the latter, neuroactive metabolites such as kynurenine and indole, leading to cancer progression. It is also well known that tryptophan metabolites induce immune cell apoptosis in glioma [87,88]. In brain metastatic tumors, Johnson et al. displayed an enrichment of distinct bacterial and viral taxa in the gut and oral microbiota in patients. Moreover, they demonstrated in vivo that depletion of gut microbiota by antibiotic administration decreased tumor growth in the brain with a change in cytokine secretion and an increase in anti-tumor T cell activity [89]. Hence, these data support the potential involvement of gut microbiota in the development of brain metastases.

## 5. Targeting Cytokines for Treatment

As a result of the involvement of these numerous signaling proteins in the development and spread of BM, it seems obvious that cytokines and chemokines are interesting therapeutic targets in the treatment of these metastases. Furthermore, in recent years several treatments have been developed (Table 1). Tocilizumab, an anti-IL-6R antibody, was evaluated in an intracardiac mouse model of lung cancer brain metastasis in order to target M2 microglia, where IL-6 plays an important role in the polarization toward M2. Tocilizumab administration once a week reduced the development on BM by 38%, which thus developed in 43% of cases compared to the control mice, where BM developed in 70% of cases [55]. BLZ945, a CSF1R inhibitor that depletes TAM, was used in an intracardiac mouse model of BCBM and revealed a down-regulation of genes implicated in cell cycle, tumor growth and invasion in isolated tumor cells. However, purified TAM-MG and TAM-MDM from treated animals demonstrated an enrichment of gene sets associated with neuro-inflammatory. This result was explained by the shift from CSF1R to Colony Stimulating Factor 2 Receptor Subunit Beta (CSF2Rb)-STAT5 downstream signaling associated with the expression of MMP14, which correlates with demyelination in neuro-inflammation, and triggers IL4-mediated wound repair mechanisms. With regard to this result, an inhibitor of STAT5, AC4-130, was added to BLZ945 and led to significant synergistic anti-tumor effects with reduced tumor growth. Moreover, the combination induced a morphological change of TAM and reduced neuro-inflammation [90]. Previously, we discussed the key role of G-CSF in the maturation and recruitment of proinflammatory neutrophils in BM. It has been shown in several in vivo models that G-CSF blockade partially inhibits or prevents the growth of BM [73,75]. Based on these observations, it would seem appropriate to consider brain-infiltrating neutrophils as a therapeutic target for the treatment of BM. An antibody capable of blocking the G-CSF receptor, CSL324, is currently in a phase I clinical trial (NCT03972280) for the treatment of inflammatory and immune disorders. However, it would be very interesting to also investigate in clinical trials the efficacy of this molecule to potentially block the G-CSF/G-CSF receptor in the treatment of BM [73]. Furthermore, neutrophils play an important role in the TME of BM and are potential targets for improving the efficacy of existing treatments. It is therefore possible to directly target TANs to inhibit their recruitment, activation or to reprogram their phenotype [74]. Indeed, the presence of neutrophils in the TME and their phenotype determine the efficacy of traditional cancer therapies such as chemotherapy but also of more recent treatments such as immunotherapy [91,92]. Therefore, recent research has focused on understanding the immune environment of brain tumors. Phase I and II clinical trials of TGFβ pathway inhibitors are underway to assess their effectiveness in promoting the development of anti-tumor neutrophils (Galunisertib NCT02672475, NCT01582269, NCT01682187; Fresolimumab NCT01401062, NCT02581787). Another target already well studied in the literature is the CXCR2 receptor. Researchers have identified that activation of this receptor by some cytokines (CXCL1, CXCL8 or IL-8) can be used by brain metastatic tumors to program tumor-infiltrating TANs into a pro-NETotic state, in order to control the distribution of cells to aid migration and invasion of metastatic tumor cells. According to an in vivo study, inhibition of CXCR2 with the antagonist AZD5069 could limit pro-tumorigenic neutrophil responses and thus delay tumor development. The CXCR2 antagonist may not only significantly decrease the influx of NETs to the brain metastasis, but also reduce the overall density of NETs produced by the infiltrating neutrophils. However, further studies are needed to investigate the ability of AZD5069 to cross the BBB [36]. Inhibition of the CXCR2 chemotactic axis would therefore induce inhibition of neutrophil recruitment to reduce inflammation in BM, and attenuate granulocytosis and vascular permeability [74]. SX-682 (NCT03161431) and Reparixin (NCT02370238, NCT05212701, NCT02001974) are two other CXCR1/2 inhibitors that are currently being studied in clinic to assess their efficacy and safety.

## 6. Discussion and Conclusions

Cytokines are crucial players that mediate key interactions between immune and non-immune cells in the TME leading to cancer progression [93]. Our review highlights the role of certain cytokines and chemokines in the establishment and development of CNS metastasis (Table 2). Some of them appear with a central role, such as IL-1β, IL-6, CCL2 and MIF. Indeed, IL-1β is an alarm cytokine that acts through IL-1 receptor to initiate and amplify local inflammation. It may also promote the production of nitric oxide and reactive oxygen species which are acting as carcinogenic mediators [93]. Interleukin-6 leads to tumor progression by the induction of several pathways, such as PI3K/AKT, mitogen-activated protein kinase (MAPK)/extracellular signal-regulated kinase (ERK) and NF-κB, which increase the expression of anti-apoptotic proteins, stimulate cancer cell proliferation and metabolism, and promote angiogenesis via the production of VEGF [93]. Concerning the chemokine CCL2 and the cytokine MIF, it was demonstrated in breast, lung and head and neck cancers that both factors are associated with cancer progression mainly via the reorganization of immune cell activities in the microenvironment [94,95,96]. Other cytokines are also observed in the serum. For example, in breast cancer, the investigation of cytokine profile by Kawaguchi et al. identified three different cytokine expression patterns between healthy volunteers and breast cancer patients and showed that metastatic patients are associated with a specific cytokine signature. The first group had higher expression of VEGF, IL-9, GM-CSF, IL-13, IL-4 and IFNγ, the second group was composed of IL-8, IL-10, IL-12, IL-5, IL-7, IL-1α, G-CSF, IL-1β and TNFα, while the last group was associated with IL-2, Eotaxin, MIP1β, MIP1α, IL-17 and bFGF. Moreover, these researchers were the first to show that serum IL-17 is upregulated in BC patients and that high levels of IL-17 correlate with a poor prognosis. Hence, the analysis of the cytokine network would provide new information for early action in the treatment of tumors and for a more effective therapeutic strategy [17]. Some molecules targeting cytokines revealed a significant impact on tumor growth and management of tumor microenvironment such as BLZ945, CSL324 and Reparixin, which are more and more being evaluated in clinical trials (Table 1). Therefore, it will be interesting to further study the profile of these inflammatory cytokines in CSF samples of brain and leptomeningeal metastases patients in order to refine the panel of cytokines involved in the development of CNS metastases leading to the discovery of new biomarkers as well as new therapeutic targets.

## Figures and Tables

**Table 1 biomedicines-10-01537-t001:** Promising treatments targeting cytokines involved in BM development.

Target	Agent	Effects on Immune Cells	References
IL-6 pathway	Tocilizumab (an anti-IL-6R antibody)	Target M2 microglia to inhibit M2 polarization to promote antitumor phenotype	[55]
CSF1 pathway	BLZ945 (a CSF1R inhibitor)	Deplete TAM, down-regulate genes implicated in cell cycle, tumor growth and invasion in tumor cells	[90]
	AC4-130 (a STAT5 inhibitor)	Use in combination with BLZ945 to have synergistic anti-tumor effects to reduce tumor growth; the combination induces a morphological change of TAM and reduces neuro-inflammation	[90]
G-CSF pathway	CSL324 (a G-CSF receptor inhibitor)	Treatment of inflammatory and immune disorders (phase I), potentially blocks the BM growth by targeting proinflammatory neutrophil recruitment	NCT03972280 [73,75]
TGFβ pathway	Galunisertib (a TGFβR1 kinase inhibitor), Fresolimumab (an anti- TGFβ antibody)	Promote the development of anti-tumor phenotype of neutrophils	NCT02672475 NCT01582269 NCT01682187 NCT01401062 NCT02581787
Chemokine signaling	AZD5069 (antagonist of CXCR2)	Limit pro-tumorigenic neutrophil responses and thus delay tumor development	[36]
	SX-682 (CXCR1/CXCR2 inhibitor) Reparixin (CXCR1/CXCR2 inhibitor)	Inhibit neutrophil recruitment, attenuate granulocytosis, neutrophil recruitment and vascular permeability	NCT03161431 NCT02370238 NCT05212701 NCT02001974

**Table 2 biomedicines-10-01537-t002:** Cytokines and chemokines involved in BM and LP development and progression.

Primary Cancer	Site of Metastasis	Cytokines	Role	References
All cancers	BM and LM	TGFβ, TNFα, IL-6	Activate transcription factors (Smads, NF-κB and Snail) during cancer cell invasion driving EMT.	[18]
	BM	IL-1β	Increase the permeability of the BBB. Switch astrocytes into a pro-tumoral phenotype that allow communication via gap junction with cancer cells.	[20,42,43]
	BM	IFNγ, TNF	Support brain metastatic cell proliferation by activating STAT1 and NF-κB signaling.	[43]
	BM	MIF, TGFα	Induce tumor progression by decreasing CD8+ T cell infiltration in the brain microenvironment.	[44]
	BM	CXCL1, TNFα, TGFα, IFNα	Induce the expression of S100A9 in cancer cells leading to an inflammatory response and radioresistance.	[48]
	BM	CXCL10	Suppress T cell infiltration.	[54]
	BM	IL-10, IL-4, IL13	May induce the development of TAM and the suppression of CD8 T cell responses.	[69]
	BM	G-CSF, CXCL 5/8/13	Recruit immunosuppressive neutrophils to the brain for metastasis growth.	[73,75,76]
	LM	IL-6, IL-8, IL-1β	Cytokine-derived macrophages that induce LCN2 expression leading to cancer cell growth.	[78]
Lung cancer	BM	CCL2	Promote cancer cell migration across the BBB.	[25]
	BM	TNFα	Enhance E-selection expression on endothelial cells of BBB, increase the adhesion of cancer cells on endothelial cells.	[26]
	BM	TGFβ1	Mediate the production of exosomes enriched with lnc-MMP2-2, which promote the destruction of tight junction leading to an increase of the BBB permeability.	[28]
	BM	↘ IL23A	Induce cell proliferation.	[38,39]
	BM	IGF-1, CCL20	Stimulate tumor cell growth and cancer cell stemness.	[53]
	BM	IL-6	Key regulator that promotes anti-inflammatory microglia via the JAK2/STAT3 signaling pathway.	[55]
Breast cancer	BM	IL-6, IL-8, CCL5, CCL2	Facilitate cancer cell transmigration across the BBB.	[22,29]
	BM	CX3CL1, CXCL13	Alteration of the BBB permeability.	[23]
	BM	EGF, CXCL8, MMP9, CXCL16, CXCL12	Increase cancer cell invasion and migration with tumor progression.	[33,34,35]
	BM	CXCL1	Stimulate the recruitment of tumor-associated neutrophils.	[36]
	BM	CCL7	Activate glial cells.	[46]
	BM	IL-6, CCL2	Increase BTB permeability.	[47]
	BM	CCL5, ↘ TNFα	Induce tumor growth.	[54]
	BM	CNTF	Activate astrocytes to induce cancer cell growth and colonization to the brain.	[50]
	BM	CXCL1, CXCL8, IL-8	Activate CXCR2 on neutrophils to induce NETs formation that promotes metastasis.	[36]
	LM	GM-CSF	Play a major role in the establishment of LM by promoting cell growth.	[79]
Melanoma	BM	CXCL10	Promote migration of melanoma BM cells.	[30]
	BM	TGFβ2	Important role for BM establishment.	[37]

## Data Availability

All data generated or analyzed during this study are included in this publication.

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
