# Peer review of "Cytokine Landscape in Central Nervous System Metastases"

_biomedicines, 2022, doi:10.3390/biomedicines10071537_

Round 1
Reviewer 1 Report
Metastatic lesions to the brain occur fairly routinely, with the overall survival in general is poor. Standard treatments including surgery, chemotherapy and radiation therapy in general are not particularly effective. Various cytokines secreted by the tumor have been shown to be effective in facilitating the growth of the metastatic tumors. It Is suggested that better understanding of the cytokine network as altered by the tumors may be helpful in developing new treatment strategies for these tumors. Most of the potential treatment options for these tumors based on the understanding and manipulation of various critical cytokines are speculative in nature. This is a nice review of the role of cytokines in the pathogenesis of metastatic tumors to the brain. There is some discussion of 'gut-brain axis' issues during brain tumor development, and this is speculative and should probably be removed.
Author Response
Thank you for your revision.
We have developed a point on treatment targeting cytokine as suggest by several reviewers and some of them appreciate the fact that we discussed the “gut-brain axis” this is why we have kept this short part in the manuscript, highlighting some evidence of the potential role of gut microbiota in brain metastasis development.
Changes are in red in the paper.
Reviewer 2 Report
Authors presented in their review CNS as one of locations for metastases. They discussed involvement of cytokines, including interleukins and chemokines, blood-brain-barrier (BBB) extravasation, interaction between cancer cells and cells from the brain including astrocytes and microglia. Furthermore, the role of the gut-brain axis on cytokine release during this process.
The authors had an good idea for writing an interesting paper, however, does not show a sufficient scope of all new knowledge, including each of the mentioned items, and the role of the immune system in a broader sense, and as such, this paper is not suitable for publication in biomedicines- MDPI. It also does not include the latest knowledge (references) on the role of astrocytes and microglia, BBB, and especially gut-microbiota. Disscusion is rather poor.
Author Response
Thank you for your revision.
To improve our manuscript and further develop discussion, we have added a point 2.1.3 on the role of other immune cells such as lymphocytes and neutrophils in brain metastasis and a point 5. “Targeting cytokines for treatment” that discuss the use of new molecules in animal model but also in clinical trials. We have also added new recent references (from 2022) in astrocyte and microglia part.
Changes are in red in paper.
Reviewer 3 Report
The manuscript entitled “Cytokine Landscape in Central Nervous System Metastases” describes the involvement of cytokines, including interleukins and chemokines, during the development of brain and leptomeningeal metastases from the epithelial-to-mesenchymal cell transition, blood-brain-barrier extravasation to the interaction between cancer cells and cells from the brain microenvironment including astrocytes and microglia. Furthermore, the role of the gut-brain axis on cytokine release during this process is also addressed.
It suggests add information on the clinical use of the cytokines blockage in the treatments, and /or the potential targets of these cytokines in the future disease therapy.
Author Response
Thank you for your revision that allow to improve our manuscript.
As you have suggested, we have added a point 5. “Targeting cytokines for treatment” that discuss the use of new molecules in animal model but also in clinical trials.
Changes are in red in the paper.
Reviewer 4 Report
This manuscript deals with the role of cytokines in the generation of brain metastasis (BM) of solid cancers. The authors focus their attention mainly, if not exclusively, on the relevance of astrocytes, microglia and peripheral blood -derived monocyte/macrophages in the localization and further growth of BM.
The message is that a complex network of cytokines can be involved in the establishment of a metastatic niche within the brain. This is dependent on the interaction with cells of the metastatic tumor microenvironment (mTME).
It appears that other leukocyte population other than monocytes are not considered relevant in this context. Is this because the lymphocytes and neutrophils do not localize in the inflammation site? The topic of metastasis of haematological malignancies is not almost considered at all. Furthermore, no attempt to consider the cytokines as a target for therapy of BM has been performed. In this present form the manuscript is limited and some of the points raised should be considered.
It seems that the first main step to establish a niche inside the brain is well analysed, but this is limited to some inflammatory cells and a deep analysis of this cytokine landscape during the evolution of metastasis or on relapse of these metastases is not performed.
Author Response
Thank you for your revision that allow us to improve our manuscript.
Lymphocytes and neutrophils are also involved in the progression of brain metastasis. We have added a point 2.1.3 “Role of other immune cells” in the manuscript. Regarding the treatment, we also added a point 5. to discuss the use of new molecules targeting cytokines in animal models and in clinical trials.
Changes are red in the paper.
Round 2
Reviewer 2 Report
The authors revised parts of the manuscript suggested by other reviewers but didn't improve the subtitles covering the BBB, the gut microbiota, and its immunologic link to CNS metastases. In addition, they didn't significantly improve the most important part -the discussion. This work is well suited to be published elsewhere, but, as I said, not in Biomedicines.
Reviewer 4 Report
The authors have improved the manuscript, adding some relevant details on cell populations involved in the production of cytokines and which cytokines can be a target of therapy.